# Structural Integrity of the Aircraft Interior Spare Parts Produced by Additive Manufacturing

**DOI:** 10.3390/polym14081538

**Published:** 2022-04-11

**Authors:** Stepans Kobenko, Didzis Dejus, Jānis Jātnieks, Dāvis Pazars, Tatjana Glaskova-Kuzmina

**Affiliations:** 1Baltic3D.eu, Braslas 22D, LV-1035 Riga, Latvia; steathl8@gmail.com (S.K.); didzis@baltic3d.eu (D.D.); janis@baltic3d.eu (J.J.); 2CENOS, Zeļļu 23, LV-1002 Riga, Latvia; davis.pazars@protonmail.com; 3Institute for Mechanics of Materials, University of Latvia, Jelgavas 3, LV-1004 Riga, Latvia

**Keywords:** structural integrity, beam model, linear-elastic material, additive manufacturing, fused deposition modeling (FDM), Ultem 9085

## Abstract

In this paper, the results obtained for the structural integrity of two real-life aircraft interior parts produced by using Ultem 9085 and the fused deposition modeling (FDM) are presented. Numerical simulation was used to perform static mechanical analysis of the class divider subjected to the case of the most critical load. By using a simple beam model, it was identified that the most efficient way of increasing the bending stiffness (required to pass the most crucial load case test) would be to increase the part’s width of the class divider. Mechanical testing of the parts was performed in vertical and horizontal load directions to supplement the numerical results. For the class divider, it was testified that the 3D-printed part would not fail under the most critical load case. For the folding table printed as a honeycomb structure, when loaded at the tip, the critical load of 900 N was acceptable, and as it was shown, there was significant potential for further optimization of the structure to either increase the maximum load or reduce the weight for any given load.

## 1. Introduction

Due to the gradual increase in air travel, airline companies keep investing in aircraft cabin interiors, also to improve the interiors of old aircraft vehicles [1]. Updating and refurbishing interior design is an efficient approach to enhance the customers’ flying experience. One of the benefits of updating aircraft interior design is the reduction in cabin weight, resulting in a lighter aircraft that burns less fuel and is more efficient to operate. This opens endless opportunities for new routes with fewer stops and greater cost savings [2,3]. Moreover, it should be mentioned that the efficiency of an aircraft and the availability of a safety margin can be greatly diminished in the case of an overweight aircraft during potential emergency conditions. Therefore, commercial transport greatly benefits by applying elaborated design and manufacturing solutions for interior parts [1,2,3,4].

More and more composites are being applied in interior aircraft applications, accomplished by modern and advanced technologies [4]. The materials used for cabin interiors must address reliability and convenience for passengers [5]. Thus, the materials used in aircraft interiors should have appropriate mechanical properties, exceptional fire, smoke, and toxicity compliance according to FAR 25.853, as well as should contribute to saving space and reduction of weight [6]. Thermoplastics having essential flame, smoke, and toxicity (FST) resistance, as well as durability and easy fabrication, such as polyetherimide (PEI), polyphenylene sulfide, and polyetheretherketone (PEEK), are arousing interest for the application in aircraft interiors [1].

Ultem 9085, which is a thermoplastic blend of PEI and PEEK, was synthesized by injection molding and successfully applied for fused deposition modeling (FDM) technologies. This blend is characterized by high chemical and thermal resistance and flame retardancy, as well as low smoke emission, allowing it to pass most tests for the fire safety regulation. The additional advantage of Ultem 9085 is its outstanding dimensional stability and strength at elevated temperatures [7]. Owing to their excellent physical properties compliant with aircraft regulations (e.g., FAR 25.853, ABD 0031, OSU 65/65 tests, and NBS smoke tests), Ultem resins are extensively applied in aircraft interior applications. Currently, Ultem resins are processed by using the FDM technique [8,9].

The purpose of this paper is to demonstrate that the replacement of traditionally applied metals with lighter materials such as polymers is feasible for several aircraft interior parts manufactured by additive manufacturing (AM), keeping the high strength to weight ratios.

All materials used in an aircraft cabin compartment must meet the applicable requirements for that specific aircraft type. Minimum requirements for aircraft are dependent on the airworthiness category of the aircraft and the standard applicable at the time when the design was first certificated, and the minimum standard is defined by the aircraft certification basis [10,11,12,13].

In this paper, two examples of aircraft spare parts (aircraft interior components) made of Ultem 9085 are analyzed, which were manufactured by using FDM. The original parts were manufactured using the current traditional methods and the alternative replacement parts were made by using FDM technology, with some minor design adjustments to meet the mechanical load requirements and pass the certification. It should be emphasized that no such/similar results for Ultem materials concerning the structural integrity of the aircraft interior spare parts produced by AM were found to be published. Thus, the originality of this research was to demonstrate that the replacement of currently used metals with polymers is feasible for non-engineering structures in the aviation sector.

## 2. Materials and Methods

### 2.1. Selection of Materials

Within the production design, the materials were mainly selected by considering the market and legal requirements, price savings, and also environmental sustainability. The materials should have appropriate flammability, mechanical properties (e.g., strength and stiffness), low weight, and good resistance to the aircraft’s environmental functional conditions, such as pressure, temperature, and humidity during the lifecycle [14,15,16,17]. Moreover, the spare part should also be compatible with finishing and joining materials and techniques [10]. Generally, the material choice is also limited to the range of conventional materials used for aircraft applications meeting specific technical/aeronautical regulations, such as appropriate fire/smoke resistance, density, etc. [7].

The certification of aircraft cabin interior structures is divided into different categories, depending on where and how the part is used. The following main tests can be distinguished for certification of aircraft interior parts: stress analysis, static mechanical tests, and flammability tests.

Depending on the functionality of the spare part, the part or a whole component will go through all or just some of the tests. The tests for different parts are defined by airworthiness standards and regulations. For the installation in the aircraft interior, the part must meet the FAR/CS 25 type certificate requirements [11,12].

Following similar principles for passing the concentrated loads (see Table 1 and Table 2), some of the installed parts must also pass the loads applied by passengers during an emergency evacuation or turbulence. These loads may arise in the aircraft’s interior parts during the flight, including emergencies as a result of pulling, pushing, standing, stepping, or sitting by passengers. The appropriate concentrated loads should be withstood by the components that could be grasped and pulled suddenly and fast by the crew and passengers.

Thus, the concentrated loads subjected to different aircraft interior parts should be considered under the design to meet the safety requirements for passengers and crew members and also to reveal the high reliability and performance of the interior parts. Safety regulations should be combined with the design considerations to meet all requirements to primarily protect the occupants of the aircraft.

The examples of acceptable values of loads ensuring the safety of passengers and the crew are provided in Table 1 and Table 2 [11]. However, it should be mentioned that the integrity of aircraft interior parts could be affected by additional factors, such as, e.g., variability in the material properties and the process of construction of the component. Additionally, for more precise analysis, the effects of environmental conditions (e.g., temperature, moisture absorption, etc.) should be considered for polymers and polymer-based composites [18,19,20].

### 2.2. Manufacturing of the Spare Parts

The Stratasys Fortus F900 machine (Stratasys, Eden Prairie, MN, USA) was used at the Baltic3D factory for the manufacturing of all spare parts. The material used was Ultem 9085 (Stratasys, Eden Prairie, MN, USA), which is a polyetherimide thermoplastic FDM material. It is characterized by an elevated strength-to-weight ratio, thermal and chemical resistance, and also complies with standards for FST characteristics of the aerospace and railway industry. The material filament was stored in a vacuum-sealed protective bag inside a canister at a temperature of 13–24 °C and humidity <60%.

The main parameters to manufacture the spare interior parts (a class divider and a folding table) are provided in Table 3. The printing directions for the class divider and folding table are shown in Figure 1, where *z* is the printing height.

The class-divider ruling dimension was 0.485 m and printed in two parts for assembly purposes. The dimensions of both parts assembled were 0.485 m × 0.140 m × 0.034 m. For the seat folding table, the existing interface points with other structures were retained. The dimensions of the seat folding table were, accordingly, 0.4065 m × 0.173 m × 0.0196 m.

### 2.3. Numerical Simulation for the Class Divider and the Seat Folding Table

Fusion 360 (F360, Autodesk, California, CA, USA) was used to perform static mechanical analysis of the class divider subjected to the case of the most critical load. A simple linear-elastic material was assumed by using the mechanical properties of Ultem 9085. The class divider was considered a thin-walled structure with a length of 0.485 m and a wall thickness of 3 mm. As a rule of thumb, at least three solid elements are recommended along the thickness direction to accurately model curvature, but with such a large feature size difference, the total number of degrees of freedom (DOF) would be inconveniently large. Therefore, by making small design adjustments and keeping the rest of the settings the same, it was possible to interpret whether a change in the design improved the mechanical performance of the structural part.

A linear-elastic material was considered to have tensile (in the *ZX* direction) elastic modulus *E* = 2.41 GPa and Poisson’s ratio ν = 0.34 [21]. F360 does not support orthotropic material models in structural simulation models, and hence a linear-elastic law is the only option. As a conservative measure, the material properties along its weakest (*ZX*) direction were taken.

The out-of-plane load was applied at the closest interface between the class divider and the plexiglass. Moreover, the only parts included in the analysis were two class-divider halves. To simplify the problem, the mechanism at the top was reduced to simplified boundary conditions, the plexiglass was removed, but the interface between the two class-divider halves was modeled as a bolted interface all along its connecting surfaces, i.e., assuming it was one piece.

The surface curvatures, i.e., the fillets around the edges of the parts, were removed as these are not necessary for the analysis but complicate mesh generation and can impact solution accuracy. Except for the plexiglass attachment points, all other holes were removed in the design because these were not significant for the analysis, requiring a super-fine mesh.

For the seat folding table, two load cases were devised to test the performance of the part under load conditions. In the first load case, a force was evenly distributed along the front edge of the test specimen (Figure 2a). For the second case, the load was concentrated at one of the corners of the test specimen (Figure 2b). For the testing purpose, the boundary conditions and the hinge slots were simplified and fixed as shown in Figure 2c.

For testing the sample and applying force in the middle, the worst-case scenario was considered, where a force of 900 N was applied. According to the material data sheet provided by Stratasys [21], the ultimate strength of Ultem 9085 was 69 MPa.

### 2.4. Mechanical Testing of the 3D-Printed Parts

To test the structural responses and compare them with the finite element results, mechanical testing of the 3D-printed parts, i.e., class divider and seat folding table, was performed in the Laboratory of Experimental Mechanics of Materials of Riga Technical University. The universal testing machine, Zwick/Roell Z600 (Zwick Roell, Ulm, Germany), was equipped with a stand specially designed for the testing of these parts (Figure 3). For the testing of both parts, the following criteria were considered: the stress was applied to the area of not more than 10 × 10 cm, the strain rate was 10 mm/min, and there were two tests for each part. The first test was carried out until the maximal vertical load of 900 N, but the second one was until the maximal horizontal load of 500 N.

## 3. Results and Discussion

Two real examples of aircraft interior parts were analyzed in this paper, which are currently used in the aircraft and have a potential for replacement using additive manufacturing technology and replacing the current metal materials with the high strength-to-weight ratio, flame-retardant, and high-performance thermoplastic Ultem 9085.

### 3.1. Class Divider

#### 3.1.1. Modeling of Mechanical Properties

The class divider, as shown in Figure 4, is an original aircraft interior part, used as a design connection point to the ceiling wall, to separate the different ticket class sections. Its full assembly consists of three distinct components, a general class-divider body, a release mechanism, and mounting screws and a plexiglass wall. The class-divider body consists of two CNC machined metal halves, which are connected.

Figure 4 shows the assembled class divider and the inside structure of the two CNC metal halves, when separated. Two metal parts and simulating loads were analyzed as they would be manufactured from Ultem 9085 material with small design adjustments to compensate for the switch from the metal to the polymer.

Judging from the part’s slender shape and long moment arm, it is reasonable to expect that the top right or bottom left load cases would be the most difficult to meet. This suggests that initial design changes should focus on the structure’s bending stiffness around the *z*-axis.

For a simple cantilever beam loaded at the tip, the expression for its bending stiffness, *K*, is: (1)$$K=\frac{2EI}{L^{3}},$$
where *E* is Young’s modulus, *I* is the second moment of area, and *L* is the length of the part (*L* = 485 mm). Although it is possible to manipulate by exploiting anisotropy induced by the additive manufacturing process, in practical terms, this term can be considered fixed since there was only one material used in this study. It is not practical to change the length of the class divider because the length of the plexiglass wall and where the load is applied on it is fixed. Changing the second moment of area, *I,* is the only option to influence the bending stiffness of the part.

Thus, assuming a hollow, rectangular cross-section of the class divider in the *XZ* direction, the second moment of the area around the *z*-axis, *I*_zz_, can be expressed as: (2)$$I_{zz}=\frac{w_{1}^{3}h_{1}-w_{2}^{3}h_{2}}{12},$$
where *w*_1_ and *h*_1_ are the width (*x*-direction) and height (*z*-direction) of the full rectangle, and *w*_2_ and *h*_2_ are the width and height of the hollow section. If the wall thickness, *t,* is constant, it can be expressed as: (3)$$t=\frac{\left(w_{1}-w_{2}\right)}{2}=\frac{\left(h_{1}-h_{2}\right)}{2}.$$

The value for *I*_zz_ can be scaled most effectively by changing the part’s width and thickness of the cross-section area. Assuming a beam model, the most efficient way of increasing the bending stiffness to pass the test would be to increase the width of the part in the *x*-direction.

The testing procedure for the class-divider part can be considered from [11], where the horizontal force load of 900 N was applied, as shown in Figure 5. The beam model is a good starting point to estimate the stress at failure. Thus, in this study case, the maximum axial stress, σ_max_, can be estimated as in Equation (4).
(4)$$\sigma_{\max}=\frac{FLw}{2I_{zz}}.$$

By the application of a force of 900 N to the class divider having the length described in Section 2.2, the second moment of the area (*I*_zz_) was 420.192 mm^4^, and the maximal stress (σ_xx_) of 31.16 MPa was achieved. For the calculation of *I*_zz_, a uniform wall thickness *t* = 3 mm was considered, and the height and width of the full rectangle, *h*_1_ and *w*_1_, were set to 70 and 60 mm, accordingly. The height and width of the hollow section, *h*_2_ and *w*_2_, were derived from Equation (3).

The stress to failure along the weakest orientation for Ultem 9085 material was 49 ± 9 MPa [21]. It means that in nearly 95% of cases (2σ confidence interval), this structure would not fail under the most critical load case. All the dimensions (besides the width) used in this example were the actual measured dimensions. The actual width was 34 mm, but 60 mm is still a real choice and could be set even higher. Assuming a stress concentration factor of 3, which is typical for circular holes, the maximum axial stress should be reduced by a factor of 3 to 16.3 MPa. For such a case, changing the width and height to 85 mm and setting the wall thickness to 3.5 mm yields a maximum axial stress of 14.66 MPa. Again, these dimensions still seem to be reasonable.

#### 3.1.2. Numerical Modeling for the Simplified Design

One half of the simplified final design iteration class divider, along with the applied out-of-plane load and locations of boundary conditions, are shown in Figure 6a. The curved boundary was modeled as a pin and the flat boundary surface was fixed in all three directions.

An in-plane view of this design is shown in Figure 6b. The width, height, and wall thickness were changed with extra material added around the pin boundary conditions as this is where a lot of stress is concentrated. The shown rib pattern is just an estimated idea of providing extra load paths between sections where the load is applied and sections that can carry the load. The rib pattern serves as an example and can be subjected to change in future design iterations.

The displacement and stress plots are presented in Figure 7. The stress plot is viewed in-plane while the displacement is viewed from the top since the deformation occurs out-of-plane. From the displacement plot, it seems that the structure is still within the small deformation regime, so the linear analysis assumption holds.

The stress plot shows that the majority of the structure is below a stress value of 12.5 MPa. There is an above-failure stress point in the proximity of the pin joint boundary. A stress concentration around this location makes intuitive sense and it could be further reinforced locally, possible with some metal inserts that could take the load. Most of the structure has a safety factor SF = 49/12.5 = 3.92, which is a good starting point for physical tests without risking premature failure.

The new proposed part with minor design modifications to adjust the change from metal to polymer structure should make up in weight to strength, but in any case, the part should be verified through actual testing. This simple model serves as a motivation that a class-divider redesign could be possible using the Ultem 9085 material. Local reinforcements and other insights might still be required, but these analytical estimates suggest that it is possible.

#### 3.1.3. Further Design Modifications

From the above calculations and with the possibilities of additive manufacturing technology, it is possible to make even further design modifications to the initial design part and include complicated design appearances that would be beneficial and enhance the customers’ flying experience. Figure 8a shows one such possible solution. In the original design with increased thickness in the x-direction and a thickened surface, to pass the applied concentrated loads, an appearance of a mountain shape was added to the parts’ surface structure. Figure 8b shows the two class-divider parts printed out using Ultem 9085 material. The final weight of the manufactured part with Ultem material is roughly 2.6 kg, which is a little more than the metallic parts, but that is because an additional design appearance was added to the design of the parts. Considering the significant difference in the density of metals, e.g., lightweight aluminum (2.7 g/cm^3^) and Ultem 9085 (1.27 g/cm^3^), by further changing the final design, there is room for potential weight savings.

Therefore, this simple, idealistic model serves as a motivation that a class-divider redesign using the Ultem 9085 material could be possible. The benefit of using additive technology, unlike CNC technology, is the variety of design possibilities that additive technology can provide, and which can be added to the final part design.

A further model redesign is possible to reduce materials’ requirements and therefore further weight reduction. Attachment points remained the same, so the attachment of the final parts to an aircraft’s interior would not change. Figure 9 shows the 3D model of the redesigned class-divider part, with reduced material and improved design, where instead of two separate parts, now the part is manufactured from one single piece. With the final design improvements, the weight of the final parts was 1.6 kg, which was 0.8 kg or 33% lighter than the CNC machined part.

#### 3.1.4. Mechanical Testing of 3D-Printed Part

The testing procedure of the 3D-printed class divider in vertical pull downward and horizontal push forward is shown in Figure 10a,b, accordingly. The resulted load-displacement curves are provided in Figure 10c. The test in vertical pull downward was carried out for the part when the maximal load of 900 N was gradually increased. It should be noted that during this test, no mechanical damages and cracks were identified for the part. The maximal displacement at the point of load application during the test was 21 mm.

The test in the horizontal push forward was performed until the maximal load of 500 N. The loss of stability for the part construction was observed during the test, starting from its end and until the point of the support. One out of three screw holders was broken, which is notable as a load drop in the load-displacement graph at approximately 250 N. Moreover, a relatively high maximal deformation of 72 mm was reached during the test. However, after the load removal, no residual plastic deformations were observed for the part, and no other cracks or damages were noticed except the breakage of one screw holder.

### 3.2. Seat Folding Table

#### 3.2.1. 3D Printing

The seat folding table used for the aircraft applications and its inside structure are shown in Figure 11. It serves as a non-structural functional part and is located on the side of the chair structure, made from multiple components that perform the function of closing and opening a special compartment for the passengers to store their items.

The folding table consists of multiple components and requires multiple manufacturing options as well as additional assembly requirements. The inside structure of the seat folding table is shown in Figure 11b. The full assembled weight is 913 g, with 7 different parts, not including all the rivets and screws, and different manufacturing technologies were used, such as CNC, sheet cutting, and vacuum forming.

The possibility to replace multiple assembled components with only one part made of Ultem 9085 material by using AM was examined. Figure 12 shows an example of a 3D-printed body assembled with hatches and latches. The full assembled weight of the 3D-printed folding table and assembled mounting points was 496 g, which was 417 g or 46% lighter than the metal one.

The inside structure of the part was made with a rectangular grid pattern, to reduce its weight and keep the part’s structural integrity. The view of the inside part of the 3D-printed structure is provided in Figure 13.

#### 3.2.2. Numerical Modeling of Mechanical Properties

The distribution of Von Mises stress under a critical load equal to 900 N as described in Section 2.3 is shown in Figure 14a, with a maximum stress of 43.18 MPa at the boundary point. For testing the sample and applying force on the edge, a force of 500 N was applied. The Von Mises stress distribution under a force of 500 N applied on the edge is shown in Figure 14b.

#### 3.2.3. Further Design Modification

Just for assumption purposes, an improved design was devised which optimized the rectangular grid pattern from the initial design. This new design, shown in Figure 15a, was devised to optimize the webbing structure for the first load case. The final weight of the new design was calculated to be 412 g, which is 17% lighter than the first proposed design.

According to Figure 15b, in the new proposed design, the maximum stress was 37.4 MPa at the boundary point, which is a reduction of 13.3% compared with the first proposed design. Even though the new proposed design is lighter, which is one of the most important aspects of the interior compartments, it is hardly manufacturable using AM technology. The main reason is that the part should be optimized for a different print orientation because it was printed within current research when the material layers were perpendicular to the loading direction, and therefore yield stress was not high. By adjusting the fibers to match the loading direction, the yield stress can be increased. Thus, to reduce additional weight, further design iterations would be required, but it does illustrate that there is room for improvement, and significant mass savings could be made.

The proposed design of the seat folding table, with minor modification, which was adopted for the additive manufacturing technology, shows that the stress distribution under the load applied withstands the material’s (Ultem 9085) ultimate strength. Since the seat folding table serves as a non-structural part, it would not require such huge loads to be applied.

#### 3.2.4. Mechanical Testing of 3D-Printed Part

The testing procedure of the 3D-printed seat folding table in vertical pull downward applied in its middle and on one side is shown in Figure 16a,b, while the resulted load-displacement curves are provided in Figure 16c. Similarly, as with the class-divider part, the test in vertical pull downward was carried out for the part when the maximal load of 900 N was gradually increased. Again, no mechanical damages were identified for the part during this testing, as well as no sounds characteristic of crack appearance were defined. Moreover, after the test, the part fully recovered its original dimensions and no residual plastic deformations were observed. It is also obvious from Figure 16c that both the load-displacement curves are rather linear, which affirms the existence of only elastic deformations.

The test in vertical pull downward applied on one side of the 3D-printed seat folding table was performed until the maximal load of 500 N. Likewise, for the loading in the middle of the part, no evident mechanical damages or cracks were identified for the part during the loading, and no residual plastic deformations were found after the removal of the load. As was mentioned before, the load-displacement curve for this loading condition was rather linear, indicating the existence of only elastic deformations within this load range.

## 4. Conclusions

The structural integrity of two different parts, two real-life examples of the aircraft applications (class divider and seat folding table), was analyzed. It was found that:-A simple, idealistic modeling approach (beam model, linear-elastic material, and static mechanical analysis) could be effectively applied to estimate the structural integrity of such complicated parts. The results obtained could be a good basis for further physical tests without risking premature failure.-For the class divider, in nearly 95% of cases (2σ confidence interval), the 3D-printed part would not fail under the most critical load case. All the dimensions (besides the width) used in this example were the actual measured dimensions. The results of mechanical testing carried out in vertical pull downward and horizontal push forward proved that the part could withstand the maximal load without evident breakage, and no residual deformations were present for it after the test.-For the seat folding table, serving as a non-engineering structural part, when loaded at the tip, it could at best withstand a load of 900 N. The results of mechanical testing performed in vertical pull downward applied in the middle and on one side of the 3D-printed seat folding table proved that the part could withstand the maximal load without evident breakage and appearance of residual deformations. Moreover, preliminary analysis showed that there is a significant potential to optimize the structure to either further increase the maximum load or reduce the weight for any given load.

For sure, local reinforcements and other insights might still be required for these real-life parts, but the numerical and experimental results obtained during this study proved that the additive manufacturing technology can replace the current manufacturing technologies, and therefore, reduce the steps required for part production, delivery, and assembly. Thus, the current metal parts can be replaced by polymer materials having high strength-to-weight ratios, and also reduce the number of components needed for the fabrication of the interior part.

Thus, additive manufacturing technology is a good application for the reduction of waste material, freedom of design, replacing current manufacturing technologies, and replacing multiple materials with a single material. Finally, by applying additive manufacturing technology principles and replacing metal parts with polymer materials, the weight of the parts is reduced, which saves on the total aircraft weight, and further decreases fuel consumption and CO_2_ release.

## Figures and Tables

**Figure 1 polymers-14-01538-f001:**
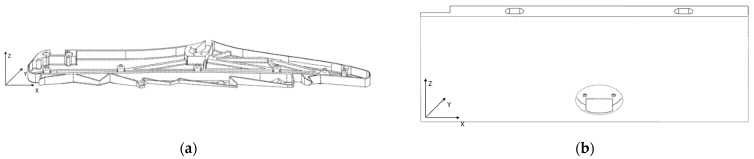
Printing directions for the class divider (**a**) and seat folding table (**b**).

**Figure 2 polymers-14-01538-f002:**
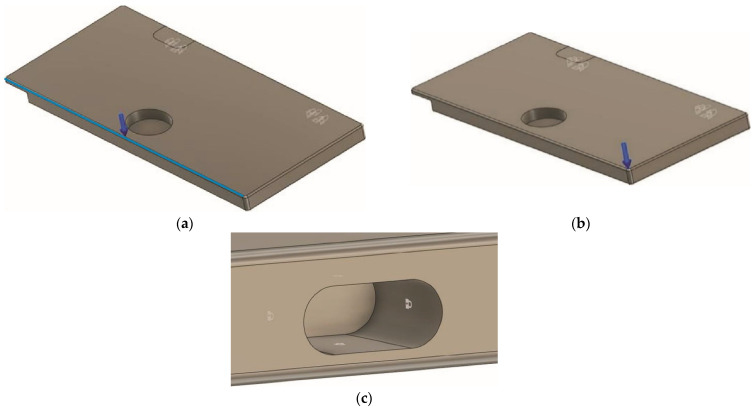
Load case 1 (**a**), 2 (**b**), and boundary conditions for the seat folding table (**c**).

**Figure 3 polymers-14-01538-f003:**
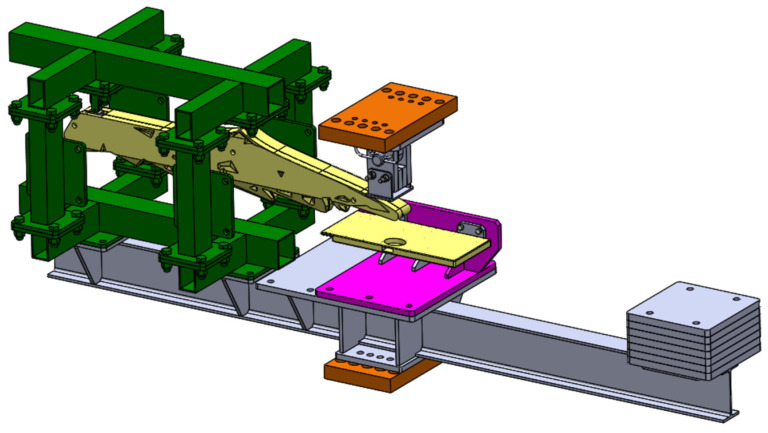
3D model for the manufactured test stand.

**Figure 4 polymers-14-01538-f004:**
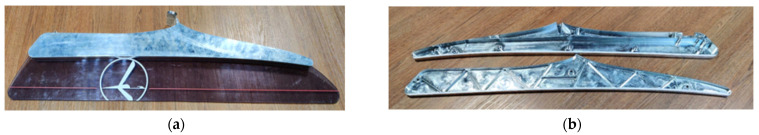
Class-divider part: (**a**) assembled, and (**b**) inside view of the two halves of the class divider.

**Figure 5 polymers-14-01538-f005:**
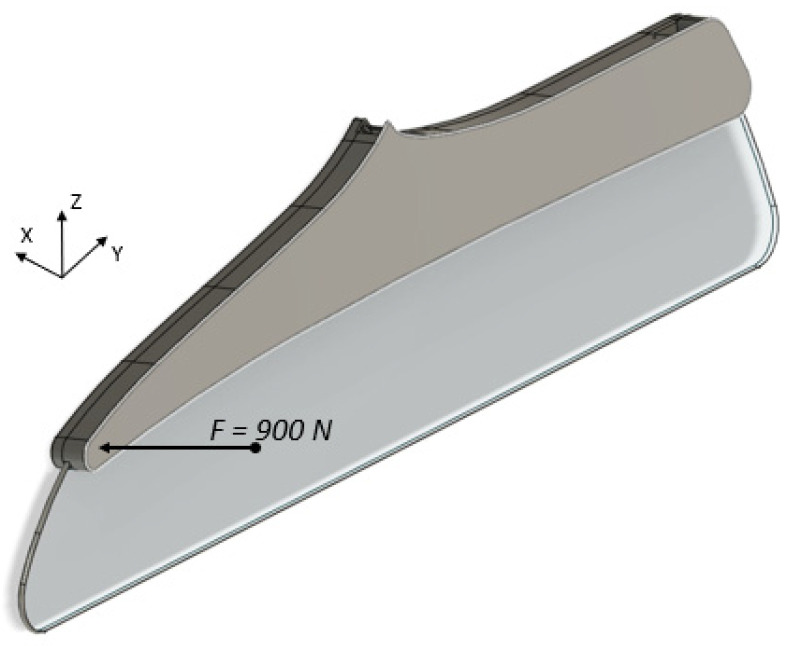
Class divider under the application of a horizontal pull.

**Figure 6 polymers-14-01538-f006:**
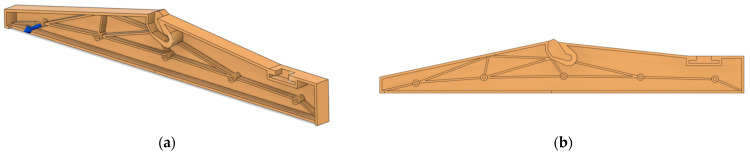
One half of the simplified class-divider model shows where the loads and boundary conditions are applied in Fusion 360 (**a**), and in-plane view of one half of the class-divider design (**b**).

**Figure 7 polymers-14-01538-f007:**
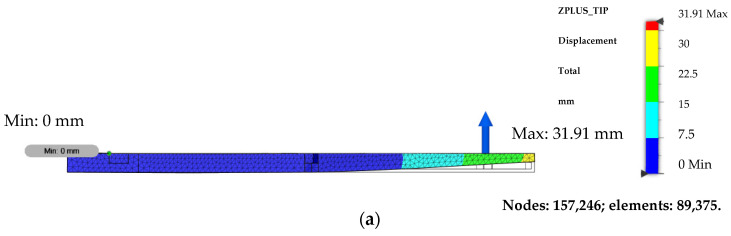
Displacement plot viewed from the top (**a**) and von Mises stress plot viewed from the front (**b**).

**Figure 8 polymers-14-01538-f008:**
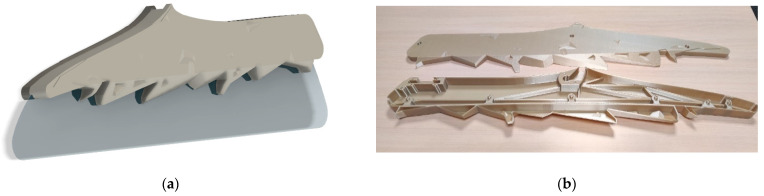
3D design model of the class-divider part (**a**) and a real-life example of a 3D-printed class divider (two parts) with the modifications to pass the applied concentrated loads and design appearance of mountains (**b**).

**Figure 9 polymers-14-01538-f009:**
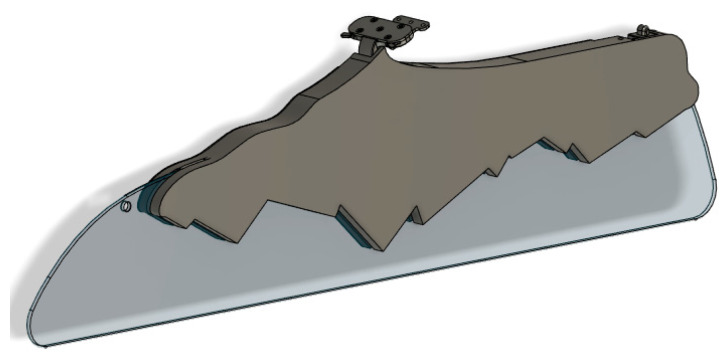
Redesigned class-divider part.

**Figure 10 polymers-14-01538-f010:**
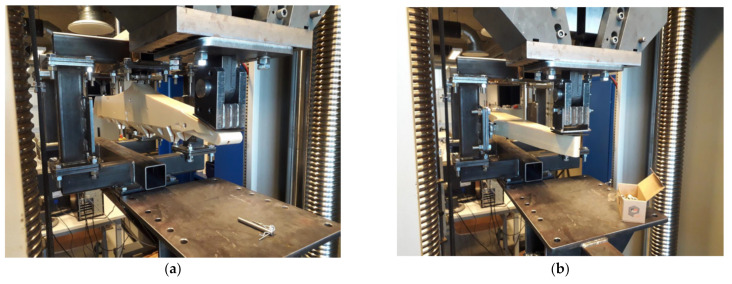
3D-printed class divider during mechanical testing in vertical pull downward (**a**) and horizontal push forward (**b**), and associated load-displacement curves for the load conditions indicated on the graph (**c**).

**Figure 11 polymers-14-01538-f011:**
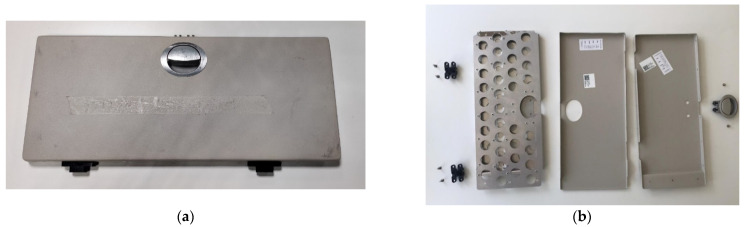
Seat folding table (**a**) and inside the structure of the seat folding table (**b**).

**Figure 12 polymers-14-01538-f012:**
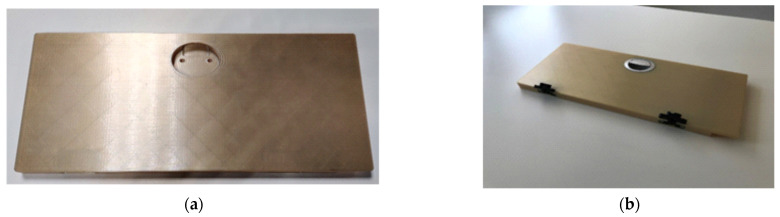
3D-printed folding table from Ultem 9085 material (**a**), and the folding table assembled with mounting points (**b**).

**Figure 13 polymers-14-01538-f013:**
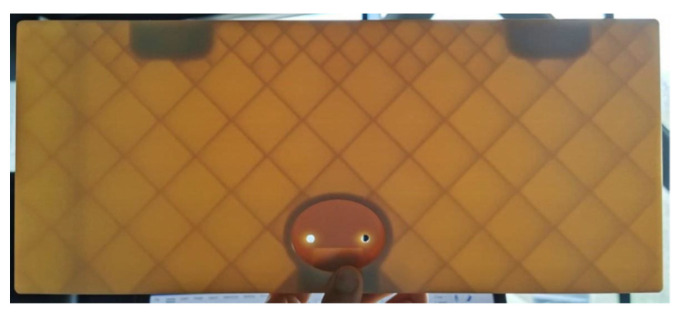
Inside view of the 3D-printed seat folding table.

**Figure 14 polymers-14-01538-f014:**
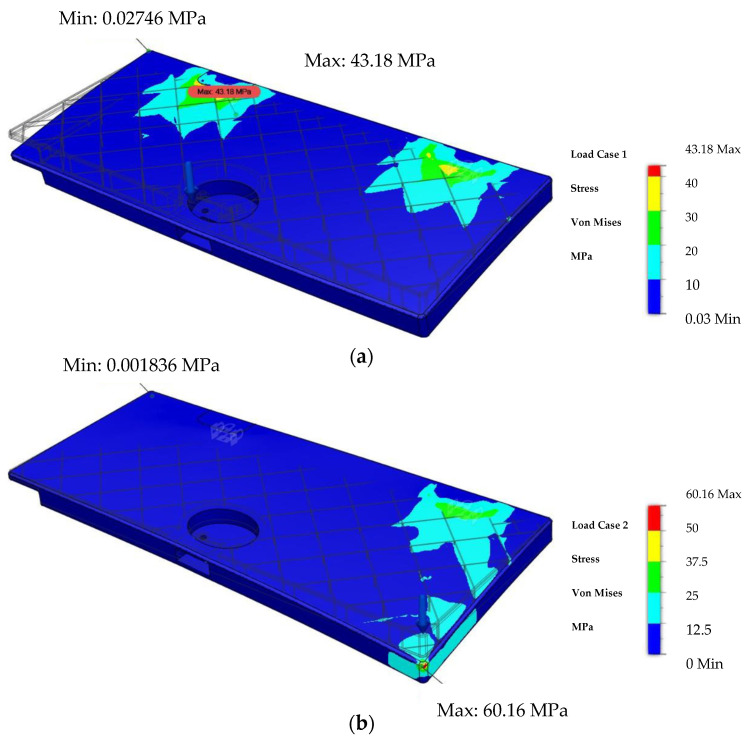
Von Mises stress distribution under the load of 900 N, subjected in the middle of the test sample (**a**), and under the load of 500 N, applied on the edge of the test sample (**b**).

**Figure 15 polymers-14-01538-f015:**
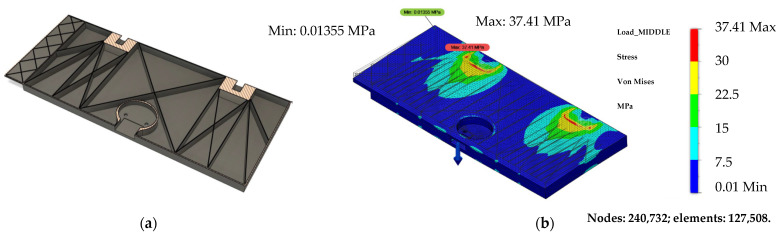
New table webbing design idea (**a**) and Von Mises stress distribution, with the load of 900 N subjected in the middle part (**b**).

**Figure 16 polymers-14-01538-f016:**
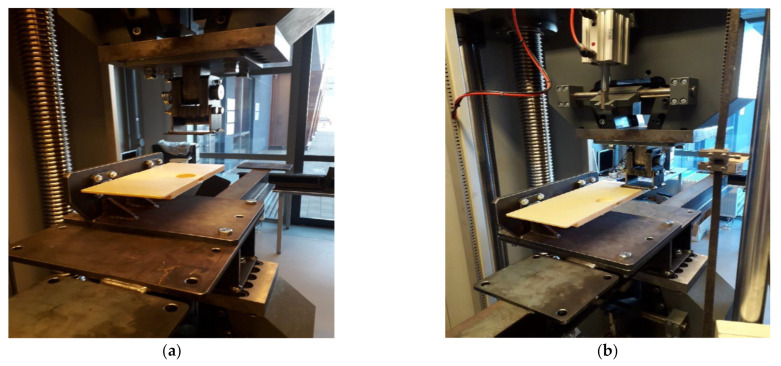
3D-printed seat folding table before mechanical testing in vertical pull downward applied in the middle (**a**) and on one side (**b**), and associated load-displacement curves for the loading conditions indicated on the graph (**c**).

**Table 1 polymers-14-01538-t001:** General concentrated loads [11].

Action	0–150 cm above the Floor, N	At 200 cm above the Floor, N	Application Area, cm^2^
Pushing	1330	440	100
Horizontal pull 1 (hand)	660	220	100
Horizontal pull 2 (hands)	660–1330	440	100
Up	660	220	100
Down	880–1330	440	100
Seating or stepping	1330–2220	N/A (up to 100 cm)	900 (seat)
			200 (step)

**Table 2 polymers-14-01538-t002:** Concentrated loads subjected to some aircraft interior parts [11].

Application Type	F, N	Comments
Partitions, galleys, lavatories	890	If used as firm handholds
Handgrip interior components	890	If used as firm handholds
Handgrip exit areas and doors	1330	Pull load
Handrail	1330	Down direction
Handrail	890	Side direction
Free span curtain track	890	Down direction, 0–200 cm above the floor

**Table 3 polymers-14-01538-t003:** Manufacturing parameters for the spare parts.

Name	Value	Units
Material tip	T16	-
Build chamber temperature	180	°C
Layer height	0.254	mm
Toolpath width contour	0.508	mm
Toolpath width infill	0.508	mm
Infill angle	45	degrees
System mode	thin wall	-
Infill density	100	%

## Data Availability

The data presented in this study are available on request from the corresponding author.

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
