# Peer review of "Structural Integrity of the Aircraft Interior Spare Parts Produced by Additive Manufacturing"

_polymers, 2022, doi:10.3390/polym14081538_

Round 1

Reviewer 1 Report

The work is concerned with designing and analysis of two parts used in aircraft, the divider and folding table. The novelty would be in the 3D printing these two parts for testing. Linear elastic analysis was performed using finite element analysis. The new products are lighter that the existing ones.

The work is interesting. However, presentation needs to be improved. The section of "results" is not so well structured and could be more logically organised. The values of the 3D printing material properties are from literature. It would be better to test them by the authors themselves. 

Finally, it would be interesting to test the structural responses and compare with the finite element results. 

Reviewer 2 Report

Dear authors

Congratulations for this interesting manuscript. The paper deals with an interesting subject precious of investigation but needs proofreading and improvement. To be accepted for publications following revisions are recommended:

Line 25: 'and also to improve' instead of 'to improve also'

Lines 27-29: One of the benefits of updating aircraft interior design is the reduction in cabin weight, resulting a lighter aircraft that burns less fuel and is more efficient to operate.

Line 42: 'arousing interest' instead of 'piquing interest'

Line 49: ', passing' instead of 'and pass'.

Line 58: specify earlier in the text that the acronym AM stands for 'additive manufacturing'.

Line 88: Delete the following snippet: ‘as well as where it is located’. Other possibility is rewrite as follows: depending on how the part is used as well as where it is located.

Line 89: delete the following snippet: 'that are performed on parts'.

Line 94: 'the part' instead of 'the interior part'

Table 1 and 2: Fix line numbering overlapping in tables

Figure 1: If possible, improve the quality of the images in figure 1.

Line 160: Explain why the class divider was considered a thin-walled structure with a length of 0.50 m and a wall thickness of 3 mm, given that its dimensions are 1.006×0.140×0.062.

Line 180: It would be very interesting to present a figure illustrating the differences between the domain before and after the simplifications adopted in the simulation of the class divisor.

Line 196: strength-to-weight ratio.

Why L = 485 mm where the dimensions of the class divider presented before are 1.006×0.140×0.062 m?

Check Equation 4. Wouldn't it be sigma_max = FLw/2I_zz? w instead of h

Line 235: 420,192 instead of 420 192

Line 237: Why h2 = h1? What is the value?

Figures 6 and 11 could be presented with a better quality

The term cosmetic was used 6 times throughout the manuscript and I believe it is not appropriate for these types of applications.

Line 288: The argument can be improved by reporting the density difference between the two materials.

The same phrase is used in lines 276 e 293: ‘Local reinforcements and...’

Lines 303 and 321: It would be interesting to present mass reduction results in percentage terms as well.

Lines 318-319 - Please, rewrite the sentence: 'one manufactured using additive manufacturing using...'

Line 329: Check the value of maximum stress. 44.8 MPa or 43.18 MPa (Figure 12)?

Lines 330-331: Rewrite the 2 sentences. Suggestion: The Von Mises stress distribution under a force of 500 N applied on the edge is shown in Figure 12b.

Line 340: ...that is, a reduction of 13,3% compared with the first proposed design.

Line 342: Why is not manufacturable using the additive manufacturing technology?

Line 370: strength-to-weight ratio

Line 372: 'good application for' instead of 'good example of'

Round 2

Reviewer 2 Report

Dear authors

How are you doing?

Congratulations for this excellent paper.

I´m completely satisfied with the manuscript reviewed.

Best regards.